# Immunomodulatory Role of *Staphylococcus aureus* in Atopic Dermatitis

**DOI:** 10.3390/pathogens11040422

**Published:** 2022-03-30

**Authors:** Ethan Jachen Chung, Chia-Hui Luo, Christina Li-Ping Thio, Ya-Jen Chang

**Affiliations:** 1Institute of Biomedical Sciences, Academia Sinica, Taipei 115024, Taiwan; drechung@ibms.sinica.edu.tw (E.J.C.); r08445113@ibms.sinica.edu.tw (C.-H.L.); christhio86@ibms.sinica.edu.tw (C.L.-P.T.); 2Graduate Institute of Microbiology, College of Medicine, National Taiwan University, Taipei 10051, Taiwan; 3Institute of Translational Medicine and New Drug Development, China Medical University, Taichung 406040, Taiwan

**Keywords:** atopic dermatitis, *Staphylococcus aureus*, IL-33, IL-36, Th2 cytokines

## Abstract

*Staphylococcus aureus* is a gram-positive bacterium commonly found on humans, and it constitutes the skin microbiota. Presence of *S. aureus* in healthy individuals usually does not pose any threat, as the human body is equipped with many mechanisms to prevent pathogen invasion and infection. However, colonization of *S. aureus* has been correlated with many healthcare-associated infections, and has been found in people with atopic diseases. In atopic dermatitis, constant fluctuations due to inflammation of the epidermal and mucosal barriers can cause structural changes and allow foreign antigens and pathogens to bypass the first line of defense of the innate system. As they persist, *S. aureus* can secrete various virulence factors to enhance their survival by host invasion and evasion mechanisms. In response, epithelial cells can release damage-associated molecular patterns, or alarmins such as TSLP, IL-25, IL-33, and chemokines, to recruit innate and adaptive immune cells to cause inflammation. Until recently, IL-36 had been found to play an important role in modulating atopic dermatitis. Secretion of IL-36 from keratinocytes can activate a Th2 independent pathway to trigger symptoms of allergic reaction resulting in clinical manifestations. This mini review aims to summarize the immunomodulatory roles of *S. aureus* virulence factors and how they contribute to the pathogenesis of atopic diseases.

## 1. Introduction

The human immune system is a complex network that employs a wide variety of machineries to protect and prepare the body for microbial challenges throughout the course of life. Although the basic principle is the same, genetic differences between individuals can significantly alter the makeup of one’s immune system and show differential responses. In other words, when exposed to the same antigen, a person with a more active immune system may trigger a heightened immune response and keep the body at a hypersensitive state. A condition referred to as “atopy” has been defined as the increased tendency to develop allergic diseases due to the buildup of immunoglobulin E (IgE) levels when exposed to even the slightest harmless antigens [1]. The etiology of atopy is unknown; however, clusters of atopy-related genes have been discussed and mapped to chromosomal locations associated with cytokine genes in immunoglobulin class switching mechanisms. Chromosomes 5q, 11q, 17q21, and 19q13 have all been reported to be associated with atopy [2,3,4,5,6]. Studies have also shown that sensitization to allergens during infancy can also affect the tendency for allergy to occur [7]. Nevertheless, exposure of allergens to epithelial cells prompts the production of alarmins to bridge adaptive immune responses. During hypersensitive responses, the majority of IL-4 secreted by type 2 innate lymphoid cells (ILC2) and or T helper (Th) 2 cells can create a Th2-dominant environment and allow IgE class switching, increased production of IgE, and IgE-mediated allergy by plasma cells [8]. IgE can bind to FcεRI on mast cells to induce degranulation and the release of histamine, proteases, prostaglandins, and leukotrienes to generate allergic symptoms. 

Atopic dermatitis (AD) is a chronic, relapsing inflammatory allergic condition of the skin that affects millions of people worldwide. Although the exact cause of AD still requires further investigation due to its heterogeneity, recent evidence has associated important risk factors underlying the severity of the disease from clinical observations and insights from animal models. Genetic studies have revealed epidermal barrier dysfunction as an important characteristic of early-stage AD. Histological examinations also revealed that an accumulation of immune cells is known to play an essential role in AD development, including macrophages, dendritic cells, T cells, eosinophils, and group 2 innate lymphoid cells (ILC2s) [9]. AD is generally considered to be driven by Th2 cytokines. However, as the disease progresses, infiltration of different subsets of T cells, including Th1, Th17, and Th22 cells can further compromise skin integrity and contribute to chronic inflammation [10]. 

Keratinocytes make up over 90% of the cells in the epidermis and contribute to protecting the body from the environment by constituting a physical and chemical barrier. Specific mechanisms on how skin integrity is maintained from cell-to-cell interactions is reviewed in detail elsewhere. As the most prominent cell type in human skin, keratinocytes serve as the first line of defense by recognizing pathogens through pattern recognition receptors (PRRs), which then trigger antimicrobial responses, including secretion of chemokines and cytokines to recruit immune cells. Keratinocytes are known to express Toll-like receptors (TLRs), NOD-like receptors (NLRs), and RIG-I-like receptors (RLRs) [11]. During bacterial infection, extracellular TLRs including TLR-1, TLR-2, TLR-4, TLR-5, and TLR-6 can recognize a range of highly conserved components such as peptidoglycan (PGN) and lipoteichoic acid (LTA) from gram-positive and lipopolysaccharide from gram-negative bacteria. Recognition of pathogens triggers downstream signaling, most notably through the MyD88 adaptor molecule, which plays a central role in activating NF-κB and MAPK cascades for the induction of inflammatory cytokines [12]. Secretion of pro-inflammatory and Th2-promoting cytokines from keratinocytes, as well as chemokines that attract members of the innate immunity including dendritic cells, macrophages, eosinophils, basophils, and mast cells, promotes bacterial clearance but also aggravates local inflammation; failure to fend off foreign invasion can lead to chronic inflammation. 

*Staphylococcus aureus*, a common inhabitant of the human microbiota, is usually found on the skin surface, in the nasal passages, and in the respiratory tract where they coexist with other microbes such as *Actinobacteria*, *Firmicutes*, *Bacteroidetes* and *Proteobacteria* [13]. However, a loss of diversity is observed in most patients during AD flare and results in staphylococcal colonization [14]. A huge part of their success comes from their ability to form biofilms which allow them to adhere to the host surface and provide resistance to antibiotics. They are also armed with many host cell-targeting toxins, either specific or nonspecific, that help them evade host defenses and facilitate tissue invasion. Researchers have identified several key virulence factors that play multiple roles in enhancing epithelial-derived cytokine production in *S. aureus*-mediated infections of epidermal and mucosal barriers, most notably staphylococcal enterotoxin B in ovalbumin (OVA)-induced allergic inflammation. To successfully colonize the host, *S. aureus* also encodes a special family of proteins called the phenol-soluble modulins (PSMs), which have a wide variety of functions, including colony spread, biofilm formation, and immune cell death [15]. Studies over the past two decades have provided important information on how colonization of *S. aureus* in patients with AD can affect clinical outcomes. This review will provide a better understanding of the role of *S. aureus* in potentiating inflammation in AD (Figure 1).

## 2. Atopic Dermatitis Murine Models for the Study of *S. aureus* Infection

Animal models for the study of the disease mechanisms of AD and the role of concurrent infection of pathogens in inflammation is essential. To find a suitable animal model that can accurately represent the complexity of AD in humans, researchers have come up with various mouse models mimicking different conditions of AD. However, there are still limitations, as no models can wholly represent the heterogeneous nature of human AD [16]. Currently available animal models utilize three different approaches to investigate the different mechanisms of AD pathogenesis: epicutaneous sensitization to common allergens, genetically engineered model with overexpression or knockdown/knockout of AD-related gene, and inbred strains that spontaneously develop AD-like phenotypes without allergen exposure [17]. 

Repeated application of common allergens to murine skin can generate a phenotype of AD. Epicutaneous sensitization of food allergens such as OVA with a combination of frequently used adjuvants including aluminum hydroxide, lipopolysaccharide (LPS), and staphylococcal enterotoxin B (SEB) [18,19] have traditionally been used to induce a type 2 response [20,21]. Sensitization to OVA can induce upregulation of mRNA levels of Th2 cytokines IL-4, IL-5, and IL-13, which are critical for IgE synthesis [22]. Other models using haptens [23] and calcipotriol [24,25,26] to induce Th2 responses have also been characterized [23]. Since *S. aureus* colonization of atopic skin has been associated with chronic inflammation in patients with AD, sensitization of the skin to *S. aureus* is also common to investigate its role in exacerbating clinical manifestations. Most researchers investigating the epicutaneous exposure of *S. aureus* in inflammation have utilized an approach involving securing a gauze pad with a certain number of bacteria diluted in PBS onto depilated dorsal skin of mice for a period of time to assess the severity of skin inflammation [27]. A modified approach investigating the role of *S. aureus* in AD skin involves inducing a mechanical injury by tape stripping to determine how barrier disruption facilitates *S. aureus* colonization and persistent infection [28].

Epidermal barrier dysfunction identified from genetic studies has linked the loss of key structural proteins, especially the filaggrin gene (*FLG*), to AD development [29]. Patients with loss-of-function mutations appeared to develop more severe AD symptoms and chronic inflammation [30]. To mimic the effects of barrier defects in human AD, mouse models were genetically engineered to have missing structural proteins lining the skin layers. Previous research has established that the loss of *FLG* exhibited enhanced allergen-induced inflammation, and in cases without inflammation, mutations were found to induce alterations in differential protein expression of kallikrein (KLK7), peptidylprolyl isomerase A (PPIA), and cofilin-1 (CFL1) [31]. Another important aspect of AD development is the expression of Th2 cytokines IL-4 [32], IL-5, and IL-13 in modulating inflammation [33]. Mice with overexpression of Th2 cytokines have also been used to elucidate their immune cell recruitment and activation functions. Transgenic expressions of IL-4 [34] and IL-13 [35] have been found to induce pruritic dermatitis. Mice with overexpression of epithelial cell-derived cytokines such as thymic stromal lymphopoietin (TSLP) [36] and IL-33 [37] also tend to develop spontaneous AD-like phenotypes.

## 3. *Staphylococcus aureus* Virulence Factors in Skin Inflammation

*S. aureus* colonization of the skin has been linked to the exacerbation of skin inflammation in diseases such as AD, and this phenomenon has been attributed to the activity of their repertoire of toxins and superantigens [38]. Aside from multiple reports on the use of staphylococcal enterotoxin B on immune cell activation, other virulence factors have also been reported to affect immune cells and induce immune responses. *S. aureus* is known to secrete an array of endotoxins and exotoxins [39]. Among them, α- and δ- hemolysin and a family of proteins called PSMs have been shown to cause skin barrier disruption and elicit AD-like inflammation. 

A study in 2014 investigated the effect of extracellular vesicle (EV)-packaged and soluble α-hemolysin in vitro and in vivo on keratinocytes. Hong et al. found that both forms can induce epithelial cell death in vitro; EV-associated α-hemolysin induced necrosis, whereas soluble α-hemolysin induced apoptosis. Both forms were also found to cause skin barrier disruption and epidermal hyperplasia in vivo, but only EV-associated α-hemolysin promoted AD-like dermal inflammation from increased infiltration of immune cells, particularly eosinophils, and from increased epidermal thickening. To identify the correlation between α-hemolysin and AD disease progression, Hong also isolated *S. aureus* from skin samples of healthy controls and AD patients and screened for α-hemolysin production. They found 91% of *S. aureus* from AD patients produced α-hemolysin compared to 33% of healthy controls. *S. aureus* collected from patients with severe AD also exhibit higher α-hemolysin production compared to mild and moderate groups [40]. 

Another known toxin secreted by *S. aureus* that has previously been reported to contribute to atopic dermatitis is δ-hemolysin. δ-hemolysin is a major virulence factor that can cause mast cell degranulation to trigger symptoms of allergy such as itch. In vitro treatment of a human mast cell line using *S. aureus* culture supernatant was found to induce release of mast cell mediators such as tryptase and lactate dehydrogenase. Also, skin colonization of δ-hemolysin-positive, but not δ-hemolysin-deficient *S. aureus* was found to induce production of IgE and IL-4. These studies suggest that δ-hemolysin, through induction of mast cell degranulation, contributes to linking *S. aureus* colonization and atopic dermatitis [41]. 

*S. aureus* PSMs are a family of protein produced in high abundance that is strictly regulated by the accessory gene regulator (agr) system. They produce four types of α-form and two types of β-form PSMs, classified according to their length. Studies have identified many characteristics of *S. aureus* PSMs, including the ability to increase survival and colonization in the host by assisting with colony spreading and biofilm formation [42,43,44]. More importantly, PSMs were shown to be the main factors of *S. aureus* secretome responsible for the induction of pro-inflammatory cytokines in human keratinocytes by pore formation resulting in membrane rupture and eventually cell death. A recent study has identified PSM α3 as the most potent form of PSMs as it upregulated a large panel of pro-inflammatory chemokines and cytokines, including CXCL1, CXCL2, CXCL3, CXCL5, CXCL8, CCL20, IL-1α, IL-1β, IL-6, IL-36γ, and TNF-α, while inducing the release of CXCL8, CCL20, TNF-α, and IL-6 [45]. Topical and basal layer stimulation of PSMs in an ex vivo model of human skin explants also triggered an intense induction of inflammatory response seen in vitro. These recent findings show that during *S. aureus* colonization, these virulence factors can induce a robust immune response during infection and could contribute to exacerbation of skin inflammation during AD pathogenesis. 

In 2017, the Microbial Genomics Section from NIH analyzed the skin microbial community of pediatric AD patients, hoping to shed light on how functional differences of staphylococcal strains may affect the course of AD. They were able to identify strain-level diversity using shotgun metagenomic sequencing and found dominant but distinct clones of *S. aureus* in patients with severe AD. To highlight the biological differences between strains of *S. aureus* isolated from patients with less and more severe AD, they developed a cutaneous colonization murine model and found that *S. aureus* isolated from more severe AD can induce higher immune responses. Topical applications of these isolates show differential levels of inflammation, but an overall increase in epidermal thickening and cutaneous infiltration of neutrophils and eosinophils, as well as different subsets of T helper cells, including Th2 and Th17 cells, were observed [46]. Interestingly, they identified a correlation between methicillin resistance and disease severity in patients; less severe patients were colonized with more methicillin-resistant strains (MRSA), while more severe patients were colonized with methicillin-sensitive strains (MSSA). Another study that recovered *S. aureus* isolates from pediatric patients also showed that MSSA colonization in AD patients was more prevalent than MRSA [47]. 

## 4. Th2 Profile in *S. aureus* Infection

As mentioned earlier, infiltration of different T cell subsets during the course of AD can dramatically increase complexity and lead to chronic AD. During acute AD, the skin is exposed to different antigens or specific pathogens that elicit various host responses, resulting in infiltration of different immune cell populations. Most seen is the secretion of Th2 promoting cytokines such as IL-25, IL-33, and TSLP, which activate ILC2s and promote differentiation of naïve T cells to type 2 helper cells by opportunistic bacteria in the skin microbiota [48,49,50]. However, in chronic AD, T cell-mediated responses are Th1 biased, as evidenced by the increase in Th1 cells and expression of Th1-associated genes [51].

In normal conditions where the skin is intact, foreign substances are unable to penetrate through the stratum corneum and reach underlying keratinocytes and Langerhans cells [52]. However, an impaired epidermal barrier in atopic dermatitis allows antigens to reach anatomical depths, making keratinocytes prone to exposure to foreign pathogens. Various surface molecules including peptidoglycan (PGN) and lipoteichoic acid (LTA) from gram-positive bacteria can be recognized by pattern-recognizing receptors (PRR) to initiate innate immune responses. Exposure to *S. aureus,* the main colonizer of atopic skin, increases production of TSLP and IL-33 in keratinocytes [53,54]. Studies have identified the TLR2-TLR6 heterodimer to be responsible for recognition of *S. aureus* surface lipoprotein, which promotes Th2-mediated inflammation through TSLP production. TSLP signaling has been shown to be critical for the development of skin inflammation, and increasing concentration induces the onset of Th2-associated inflammation [55]. 

IL-33 expression has been found to be upregulated in the skin of patients with AD. Researchers found that IL-33 is sufficient for the development of AD-like symptoms and the role of IL-33 in promoting Th2 response has been thoroughly explored [56]. In brief, IL-33 recruits and activates ST2-expressing ILC2s, T cells, macrophages, and eosinophils to produce Th2 cytokines, which trigger naïve CD4^+^ T cell differentiation into Th2 effector cells and promote a Th2-skewed response [57]. In vitro and in vivo findings show that *S. aureus* infection in the skin could induce upregulation of IL-33 mRNA and protein expressions. However, the reason why this bacterium can promote atopy through the IL-33/ST2 axis remains to be elucidated. Nevertheless, Al Kindi et al. recently identified an additional function of the second immunoglobulin-binding protein (Sbi) of *S. aureus* to directly stimulate IL-33 production in keratinocytes [54]. They claim that this is the first time a pathogen-specific molecule has been associated with rapid release of IL-33 in a manner that is independent of cell death. Sbi is a conserved protein that contains two N-terminal domains that can bind to the Fc region of IgG and complement factor C3b and complement factor H [58]. Knowing this, the authors inferred that it is possible that the immunoglobulin-like domains could be recognized by surface receptors expressed on keratinocytes to induce IL-33 release. Aside from its immune evasion properties, Sbi has also been shown to induce IL-6 and TNF-α in murine macrophages [59]. Though its role in the inhibition of innate and adaptive immune responses to promote bacterial survival has been characterized, this newly discovered function of Sbi on the skin and its Th2-promoting activity require further investigation to determine their clinical implication in *S. aureus* infection.

Aside from TSLP and IL-33, IL-25, a potent Th2 inducer, also plays a role in AD pathogenesis, however not much literature has investigated its effect during *S. aureus* colonization in contribution to AD exacerbation. From human patients, researchers have observed increased expression of IL-25^+^ keratinocytes and IL-17Rh1 (IL-25 receptor) infiltrating cells in AD-lesions, as compared to healthy skin. Further studies identified the importance of IL-25 in contributing to inflammation by showing that systemic IL-25 administration or transgenic overexpression of IL-25 can result in multi-organ inflammation, due to increased production of Th2 cytokines: IL-4, IL-5, and IL-13. IL-25 is produced by many cells, including epithelial cells, macrophages, eosinophils, mast cells, and basophils; however, a study focusing on acute and chronic allergic skin inflammation has identified keratinocyte-derived IL-25 as the dominant source of Il-25 for the upregulation of *il-13.* Not only does IL-25 play a part in AD-pathogenesis by inducing Th2 differentiation, but ex vivo studies have shown that IL-25 stimulation can directly lower filaggrin synthesis in keratinocytes [60,61,62,63].

Beyond Th2, a new subset of T cells expressing IL-22 were identified to work together in upregulating pro-inflammatory cytokines and chemokines in AD. Transcript levels of IL-22 and IL-22-expressing CD4^+^ and CD8^+^ T cells were found to be increased in the skin of AD patients [64]. A study investigating the effect of staphylococcal enterotoxin A (SEA) and enterotoxin B (SEB) on IL-22 showed differential modulation, in which opposite effects were observed in suppression of CD4 and enhancement of CD8^+^ T cells [65]. 

## 5. New Role of IL-36 Identified in Atopic Dermatitis

IL-36 is a newly discovered member of the IL-1 cytokine family that has recently been reported to play a role in AD development [66]. Over the past decade, researchers have neglected the role of IL-36 in mediating AD because the expression levels in the lesioned skin of AD patients were not significantly higher than those of healthy skin. Research was mostly focused on other chronic inflammatory and autoimmune diseases, as well as the role of IL-36 in colorectal cancer [67]. In 2017, Liu et al. reported that epicutaneous *S. aureus* exposure to mouse skin induced IL-36 production and promoted MyD88-dependent skin inflammation. IL-36R/MyD88 signaling induces T cell production of IL-17 to drive skin inflammation [27,68]. By comparing inflammation levels between exposure to different *S. aureus* toxin mutants, they found that this phenomenon is mediated by *S. aureus* PSMs but not α-toxin or ẟ-toxin. 

Numerous scholars have conducted extensive research and concluded that only keratinocyte-derived IL-33 and TSLP contribute to type 2 immune responses that mediate IL-4-dependent IgE production. However, Patrick et al. recently found that IL-36α produced in the inflamed skin following *S. aureus* epicutaneous exposure has direct functional activity in triggering B cell production of IgE independent of Th2 cells. Mice with impaired IL-36R activity exhibited lower serum IgE and IgG1 levels and lower inflammation levels after *S. aureus* exposure. IL-36 signaling was found to regulate B cell class switching and plasma cell differentiation by increasing transcriptional levels of *Aicda, Blimp1,* and *Xbp1* [69]. Also, IL-36α enhanced IL-4-mediated IgE production and plasma cell differentiation. Overall, these discoveries on keratinocyte-derived IL-36α provide an additional insight, beyond current literature, that shows increased IgE beyond Th2 and IL-4 contributions. 

## 6. Conclusions

AD is the manifestation of a hypersensitive reaction mediated by IgE in the skin. It continues to affect millions of people worldwide and researchers are still looking for the disease mechanism and therapeutic approaches to alleviate AD. *S. aureus* is frequently isolated and found to colonize skin of patients with AD. Colonization of *S. aureus* causes dysbiosis of the skin and has been found to exacerbate skin inflammation by the repertoire of toxins that may advance AD progression. It has been established that exposure of the skin to *S. aureus* can boost the release of epithelial cell-derived cytokines TSLP and IL-33 to promote Th2 cytokines from basophils, eosinophils, mast cells, ILC2, and Th2 cells, resulting in the buildup of IgE levels. In specific individuals with the tendency to develop atopy, these cytokines can create a Th2-dominant environment commonly seen in AD immunopathology. Recently, IL-36α has been identified as a new player that promotes IgE buildup after *S. aureus* exposure through a Th2-independent pathway, although its mechanism of action requires further investigation. 

## Figures and Tables

**Figure 1 pathogens-11-00422-f001:**
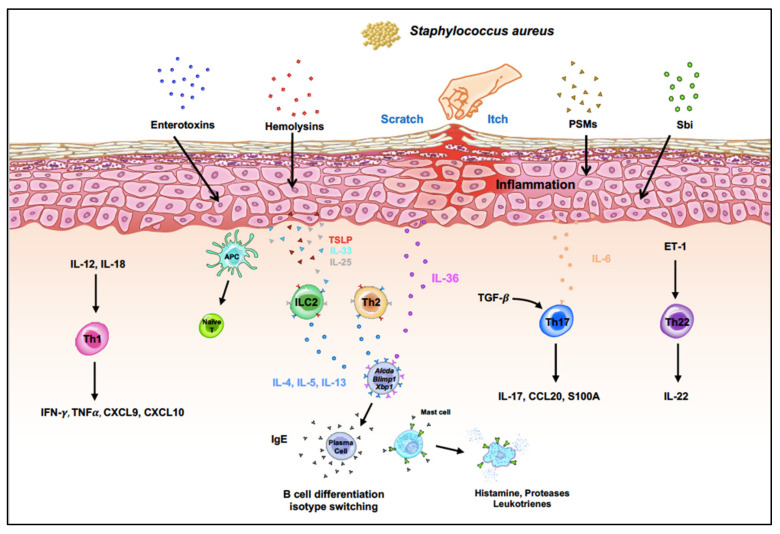
The immune profile of atopic dermatitis. Patients with atopic dermatitis are prone to infection due to fluctuations in the skin. *Staphylococcus aureus* can induce various cytokine production to generate Th1, Th2, Th17, and Th22 responses from keratinocytes and from infiltrating innate and adaptive immune cells.

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
