# Peer review of "Immunomodulatory Role of Staphylococcus aureus in Atopic Dermatitis"

_pathogens, 2022, doi:10.3390/pathogens11040422_

Round 1

Reviewer 1 Report

This review (Manuscript ID: pathogens-1612649) summarizes the updated findings on the association between Staphylococcus aureus-derived agents and immunological aspects of atopic dermatitis. The contents are generally appropriate, but the following description needs to be corrected.

1) In line 171, the phrase "proteins including PGN and LTA" is incorrect, because these substances are not generally recognized as proteins. The term "agents" is more appropriate.

2) The phrase "TLR3-TLR6 heterodimer" in lines 174-175 is a mistake for "TLR2-TLR6 heterodimer".

Minor point

3) There is a description of IL-25 in the abstract, but IL-25 does not appear at all in the main text. It is more appropriate to describe it in the relevant part of the text.

4) cyclophilin A (PPIA) → peptidylprolyl isomerase A (PPIA) or cyclophilin A (CypA) would be better.

5) immunomod-ulatory (line 23) → immunomodulatory

Remove unnecessary hyphen.

6) δ- hemolysin (line 139, line 143, line 144) → δ-hemolysin

Remove unnecessary spaces.

7) Staphylococcus aureus (line 73, Line 117) → S. aureus etc.

Check the use of other abbreviations.

Author Response

1) In line 171, the phrase "proteins including PGN and LTA" is incorrect, because these substances are not generally recognized as proteins. The term "agents" is more appropriate.

Instead of only changing "proteins" to agents, I think it would be better to refer to them as surface molecules as in gram positive bacteria, they are outside of the plasma membrane. 

2) The phrase "TLR3-TLR6 heterodimer" in lines 174-175 is a mistake for "TLR2-TLR6 heterodimer".

Thank you for pointing this out. This was indeed a typo

Minor point

3) There is a description of IL-25 in the abstract, but IL-25 does not appear at all in the main text. It is more appropriate to describe it in the relevant part of the text.

A paragraph (line 274-330) on the role of IL-25 in AD is added  

4) cyclophilin A (PPIA) → peptidylprolyl isomerase A (PPIA) or cyclophilin A (CypA) would be better.

Thank you for point this out. 

5) immunomod-ulatory (line 23) → immunomodulatory

Remove unnecessary hyphen.

removed

6) δ- hemolysin (line 139, line 143, line 144) → δ-hemolysin

Remove unnecessary spaces.

removed

7) Staphylococcus aureus (line 73, Line 117) → S. aureus etc.

Check the use of other abbreviations.

Reviewer 2 Report

This manuscript needs extensive revision as follows:

1.The authors do not present the figure summarizing how Staphylococcus aureus acts on keratinocytes, induces type 2 responses, and promotes atopic dermatitis (AD)-related pathways. Such figure is essential in this article.

2.If the authors stress the significance of Sta. aureus in the pathogenesis of AD, they should discuss if antibiotics against Sta. aureus are needed for the treatment of AD or not.

3. The authors should discuss if the Sta. aureus strains in AD patients are different from those in non-AD patients, and the different aspects of the strains, such as adherence to the skin. Further, if such difference can contribute to the differential diagnosis between AD and non-AD should be discussed.

4.The authors should discuss how Sta. aureus contribute to Th1, Th22, and Th17 profiles in AD patients.

5. Please discuss if IL-25 may contribute to the pathogenesis of AD as well as IL-33 and TSLP. 

Author Response

1.The authors do not present the figure summarizing how Staphylococcus aureus acts on keratinocytes, induces type 2 responses, and promotes atopic dermatitis (AD)-related pathways. Such figure is essential in this article.

Instead of providing a graphical abstract, we incorporated more detail into the illustration and presented it as figure 1. We also added a couple sentences from line 60 - 81 on some description of innate responses of keratinocytes.

2.If the authors stress the significance of Sta. aureus in the pathogenesis of AD, they should discuss if antibiotics against Sta. aureus are needed for the treatment of AD or not. 

Yes, S aureus colonization seem to contribute to chronic AD and AD flares. However, antibiotics are not routinely used to treat AD inflammation but only obvious bacterial infection.

3. The authors should discuss if the Sta. aureus strains in AD patients are different from those in non-AD patients, and the different aspects of the strains, such as adherence to the skin. Further, if such difference can contribute to the differential diagnosis between AD and non-AD should be discussed.

We have found a couple papers on sequencing of S. aureus strains in AD patients and have summarized relevant information from those studies from lines 217 - 239

4.The authors should discuss how Sta. aureus contribute to Th1, Th22, and Th17 profiles in AD patients.

Because  AD is Th2 driven, not much studies focused on the role of S. aureus on Th1 and Th17. Th17 cells play a bigger role in psoriasis, another inflammatory skin condition.

5. Please discuss if IL-25 may contribute to the pathogenesis of AD as well as IL-33 and TSLP. 

We have included a paragraph on the role of IL-25 from lines 276 - 334

Round 2

Reviewer 2 Report

On line 220, the authors too abruptly present the contribution of Sbi to the pathogenesis of Sta. aureus-induced exacerbation of atopic dermatitis. Regarding Sbi, the authors should firstly explain that Sbi is the component of Sta. aureus, and secondly, whether the structure of Sbi  in Sta. aureus inhabited in AD lesions is different from that in normal skin or not. 

The authors referred to 'mutant defective in Sbi'; is the mutant is specifically present in AD lesions?

These points should be explained in more detail.

Author Response

On line 220, the authors too abruptly present the contribution of Sbi to the pathogenesis of Sta. aureus-induced exacerbation of atopic dermatitis. Regarding Sbi, the authors should firstly explain that Sbi is the component of Sta. aureus, and secondly, whether the structure of Sbi  in Sta. aureus inhabited in AD lesions is different from that in normal skin or not. 

From lines 258 to 426, We have re-structured the whole paragraph on the importance of IL-33 in AD and how S. aureus infection of the skin affects IL-33 specifically. The IL-33-inducing effect of Sbi was newly discovered and no one has looked into how and why it can trigger IL-33 release. Further research has to be conducted to verify whether Sbi itself can trigger AD-like inflammation and immune responses. We also included some known research on Sbi

The authors referred to 'mutant defective in Sbi'; is the mutant is specifically present in AD lesions?

Sorry for the confusion. This was actually referring to the immune evasion function of Sbi. Researchers validated the function of Sbi by comparing phenotypes with other immune evasion protein defects and found some similarity. Regarding Sbi in AD lesions, as mentioned, no one has looked at Sbi profile in the S. aureus strains from AD lesions.

These points should be explained in more detail.